# Use of Early Donated COVID-19 Convalescent Plasma Is Optimal to Preserve the Integrity of Lymphatic Endothelial Cells

**DOI:** 10.3390/ph15030365

**Published:** 2022-03-17

**Authors:** Nada Amri, Rémi Bégin, Nolwenn Tessier, Laurent Vachon, Louis Villeneuve, Philippe Bégin, Renée Bazin, Lionel Loubaki, Catherine Martel

**Affiliations:** 1Faculty of Medicine, Université de Montréal, Pavillon Roger-Gaudry, 2900 Edouard Montpetit Blvd, Montreal, QC H3T 1J4, Canada; nada.amri@umontreal.ca (N.A.); remi.begin@umontreal.ca (R.B.); nolwenn.tessier@icm-mhi.org (N.T.); vachon.laurent@outlook.com (L.V.); 2Montreal Heart Institute, 5000 Belanger Street, Montreal, QC H1T 1C8, Canada; louis.villeneuve@icm-mhi.org; 3Department of Pediatrics, CHU Sainte-Justine, 3175 Chem. de la Côte-Sainte-Catherine, Montreal, QC H3T 1C5, Canada; philippe.begin.med@ssss.gouv.qc.ca; 4Department of Medicine, Centre Hospitalier de l’Université de Montréal, 900 Rue Saint-Denis, Montreal, QC H2X 0A9, Canada; 5Medical Affairs and Innovation, Héma-Québec, 1070 Avenue des Sciences-de-la-Vie, Québec, QC G1V 5C3, Canada; renee.bazin@hema-quebec.qc.ca (R.B.); lionel.loubaki@hema-quebec.qc.ca (L.L.)

**Keywords:** COVID-19, lymphatics, convalescent plasma therapy, endothelium, extracellular vesicles

## Abstract

Convalescent plasma therapy (CPT) has gained significant attention since the onset of the coronavirus disease 2019 (COVID-19) pandemic. However, clinical trials designed to study the efficacy of CPT based on antibody concentrations were inconclusive. Lymphatic transport is at the interplay between the immune response and the resolution of inflammation from peripheral tissues, including the artery wall. As vascular complications are a key pathogenic mechanism in COVID-19, leading to inflammation and multiple organ failure, we believe that sustaining lymphatic vessel function should be considered to define optimal CPT. We herein sought to determine what specific COVID-19 convalescent plasma (CCP) characteristics should be considered to limit inflammation-driven lymphatic endothelial cells (LEC) dysfunction. CCP donated 16 to 100 days after the last day of symptoms was characterized and incubated on inflammation-elicited adult human dermal LEC (aHDLEC). Plasma analysis revealed that late donation correlates with higher concentration of circulating pro-inflammatory cytokines. Conversely, extracellular vesicles (EVs) derived from LEC are more abundant in early donated plasma (r = −0.413, *p* = 0.004). Thus, secretion of LEC-EVs by an impaired endothelium could be an alarm signal that instigate the self-defense of peripheral lymphatic vessels against an excessive inflammation. Indeed, in vitro experiments suggest that CCP obtained rapidly following the onset of symptoms does not damage the aHDLEC junctions as much as late-donated plasma. We identified a particular signature of CCP that would counteract the effects of an excessive inflammation on the lymphatic endothelium. Accordingly, an easy and efficient selection of convalescent plasma based on time of donation would be essential to promote the preservation of the lymphatic and immune system of infected patients.

## 1. Introduction

To date, an alarming number of people worldwide have been infected with severe acute respiratory syndrome coronavirus 2 (SARS-CoV-2), a coronavirus responsible for the disease commonly referred to as coronavirus disease 2019 (COVID-19) [1]. Although many individuals with COVID-19 are asymptomatic or suffer from mild symptoms, other individuals develop severe complications, such as acute respiratory distress syndrome, disseminated intravascular coagulation and cardiovascular diseases including acute coronary syndrome and myocardial infarction [2]. A common thread between these complications and causes of mortality is the presence of impaired endothelial function [3,4,5,6]. The underlying cause appears to be the prolonged overactivation of the immune system resulting in a hyperinflammatory state, also known as “cytokine storm” [4].

Early in the pandemic, COVID-19 convalescent plasma (CCP) was proposed as a potential therapeutic option for severe COVID-19. The approach consists of transfusing plasma from donors who recently recovered from the disease to hospitalized patients, and thus transfer humoral protection against the virus. The approach was very compelling, because it was readily available. Early in the pandemic, CCP was made available in the Unites States through the Food and Drugs Administration (FDA) Expanded Access program (EAP), and later through Emergency Use Access. Initial data from the EAP suggested that passive immunization was very effective, because patients receiving CCP with high anti-COVID-19 antibody titers were 30% less likely to die from the disease than those receiving CCP with low titers [7].

The enthusiasm for CCP plummeted in winter 2021 after several major randomized clinical trials failed to show a benefit over standard of care [8,9,10,11]. The only way to reconcile a potential benefit in specific subgroups with this overall lack of effect would be if CCP was harmful in other subgroups. In fact, results from the CONCOR-1 study suggest that the previously observed relative difference in mortality with high- vs. low-titer CCP may be explained in absolute terms by a deleterious effect of CCP with low antibody titer [10]. This has important implications for the interpretation of an even more recent trial showing a benefit from CCP that used non-convalescent plasma as a control [12]. Ultimately, these results go to show how little we understand the true mechanisms underlying the immunomodulatory effect of CCP in COVID-19, which is likely not determined solely by the antibody content. This highlights the need to investigate the presence of other modulators within CCP to limit endothelial dysfunction and subsequent inflammatory complications seen in severely hospitalized patients.

Lymphatic transport is at the interplay between the immune response and the resolution of inflammation from peripheral tissues, including the artery wall [13]. Constituting initial vessels, also called lymphatic capillaries, in the peripheral tissues that drain excess interstitial fluids [14], the lymphatic system is a key player in the clearance of immune cells and pro-inflammatory modulators [15]. As vascular complications are a key pathogenic mechanism in COVID-19, leading to inflammation and multiple organ failure, we hypothesize that part of the clinical effect of CCP could derive from its effect on lymphatic vessel (LV) function. This study seeks to identify a particular signature of CCP that would decrease the effects of an excessive inflammation on the lymphatic endothelium. Following the analysis of 49 CCP donations, we observed a heterogenous effect on adult human dermal lymphatic endothelial cells (aHDLEC) in culture. We herein demonstrate that an early donation of convalescent plasma could protect against the loss of lymphatic endothelial cells (LEC) integrity observed during inflammation.

## 2. Results

### 2.1. Characteristics of COVID-19 Convalescent Plasma Donors

Forty-five donors were recruited in this study and their anthropometric data are presented in Table 1. The average age of the donors included in our study was 41-year-old (Table 1). Of the CCP donors, 38% were women. Half of the individuals experienced symptoms ranging from 10 to 20 days. Furthermore, CCP donation was made on average 67 days after the onset of symptoms. Since some studies suggest a correlation between blood type and COVID-19 outcomes [16], we characterized the blood type of donors. Forty percent of individuals were a blood type O, whereas 33%, 11% and 16% were of blood type A, B and AB, respectively. Of all 45 patients, only 7 had a negative rhesus. Most individuals had mild-to-moderate symptoms, whereas 1 and 7 people experienced no symptoms or severe symptoms, respectively. However, none of these donors were hospitalized. Individuals who lost their sense of smell or taste represented 44% of the donors. The optical density of the concentration of receptor-binding domain (RBD) of SARS-CoV-2 antibodies within CCP was on average 1.08 (minimum = 0.26, maximum = 2.41). Since dyslipidemia increases the risk to develop severe outcomes from the COVID-19 infection, we measured the level of total cholesterol of donors [17]. The average total cholesterol amongst donors was 188.5 mg/dL.

### 2.2. Elevated Antibody Concentrations and Prolonged Symptoms Are Detrimental for the Lymphatic Endothelium Integrity

Several studies suggest that the effect of CCP is dependent on its antibody titers [10,18], hence the importance of proper donor selection. The concentration of antibodies is known to increase during the first week of symptoms [19]. Since our CCP donors reported experiencing symptoms for a duration ranging between 0 and 50 days, we wanted to assess whether the concentration of antibodies was associated with the duration of symptoms. Indeed, prolonged symptoms were positively correlated with an elevation of antibody concentrations contained in convalescent plasma (r = 0.321, *p* = 0.041) (Figure 1A). We then sought to investigate whether the duration of symptoms of donors was associated with an effect on a lymphatic endothelium (Figure 1B). To measure the latter, we marked aHDLEC with the MitoSOX^TM^ Red probe following incubation of CCP to assess the production of mitochondrial superoxide, an indicator of intracellular oxidative stress. Low-to-moderate levels of mitochondrial superoxide can regulate many essential cellular processes, including gene expression and signal transduction [20]. In contrast, overproduction of mitochondrial superoxide, shown by a high expression of MitoSOX^TM^ Red (shift toward the right on flow cytometry plots of Appendix A), can trigger cellular oxidative damage that contributes to the pathogenesis of a wide variety of diseases [21]. Detection of intracellular mitochondrial superoxide is therefore of importance for understanding proper cellular redox regulation and the impact of its deregulation on lymphatic function. We found that duration of symptoms correlated negatively with MitoSOX^TM^ Red negative cells (r = −0.552, *p* = 0.018) (Figure 1B). In other words, plasma from donors with short duration of symptoms triggered a lower oxidative response in the lymphatic endothelium than that from patients who experienced symptoms for a longer duration. This highlights a potential risk of deleterious effect of plasma from a donor with prolonged symptoms on the lymphatic endothelium of recipients. Since prolonged symptoms is also correlated with antibody concentrations, we investigated whether concentration of antibodies was also associated with a detrimental effect on the lymphatic endothelium. We observed a positive correlation between the concentration of antibodies and the permeability of the lymphatic endothelium (r = 0.611, *p* = 0.035) (Figure 1C). However, by multivariate analysis, we observed the collinearity of concentration of antibodies and the duration of symptoms, indicating that they are not independent factors. Taken together, our data suggest that both the elevated antibody concentrations and the prolonged symptoms are detrimental for the lymphatic endothelium.

### 2.3. COVID-19 Convalescent Plasma from Patients Experiencing Severe Symptoms Induces Cellular Necrosis

Casadevall et al. recommended the use of CCP from donors with severe symptoms, as they have more antibodies than donors with mild symptoms [22]. Therefore, we sought to investigate whether this other parameter, namely the severity of the symptoms, could be a potential characteristic to take into consideration for the selection of convalescent plasma based on its effect on LEC. We first reported a positive correlation between prolonged symptoms and their severity (r = 0.489, *p* = 0.007) (Figure 2A). So far, studies found that older individuals experienced more severe symptoms than younger individuals [23]. This correlation was also found in our study (r = 0.560, *p* = 0.002) (Figure 2B). Furthermore, there are higher level of antibodies within CCP of severely infected individuals [22,24,25]. Similarly, a higher concentration of antibodies was also present in the plasma of donors displaying more severe symptoms (r = 0.451, *p* = 0.010) (Figure 2C). In the multivariate model (r = 0. 531), severity of symptoms was the only variable independently associated with anti-SARS-CoV-2 antibody levels (β = 0.585, *p* = 0.013). In line with previous literature, the model could be further improved (r = 0.590) by including both symptom severity (β = 0.541, *p* = 0.001) and delay since the onset of symptoms (β = −0.259, *p* = 0.09), although the latter did not reach statistical significance.

Given these results, and the detrimental effect of prolonged symptoms and higher antibody levels on the lymphatic endothelium, we sought to determine whether severe symptoms also correlated with an altered lymphatic endothelium integrity. Our results suggest that plasma isolated from patients experiencing severe symptoms is more likely to promote aHDLEC late apoptosis or necrosis (r = 0.521, *p* = 0.049) (Figure 2D) while abrogating early apoptosis (r = −0.532, *p* = 0.043) (Figure 2E). Herein, we demonstrate that while severity of the symptoms may be a good predictor of higher antibody content, it may also be associated with a plasma phenotype that could exert a deleterious effect onto the lymphatic endothelium.

### 2.4. An Early Donation Could Help Maintain the Integrity of the Lymphatic Endothelium

We then sought to determine whether another easily identifiable factor could be considered for a judicious selection of convalescent plasma. Antibody levels, especially IgM and IgA, decrease with time in CCP donated long after the onset of symptoms [26,27]. Thus, we sought to investigate the association between the time of donation and CCP effect on the integrity of lymphatic endothelial cells. It has been reported that lymphatic endothelial cell activation could contribute to help limit inflammatory diseases such as acute skin inflammation [28] and edema [29]. We herein observed that the expression of markers of endothelial activation correlated with cellular viability and mitosis and correlated inversely with lymphatic permeability and production of mitochondrial reactive oxygen species (ROS) (Appendix A). Given the potential benefits of endothelium activation markers on the integrity of the recipient endothelium (Appendix A), we aimed to evaluate whether the timing of convalescent plasma donation could impact the expression of endothelial cell activation markers in vitro. We noted a negative correlation between donation time and mRNA expression of E-selectin (*SELE)* (r = −0.486, *p* = 0.012) and of intercellular adhesion molecule 1 (*ICAM1)* (r = −0.509, *p* = 0.016) (Figure 3A,B). Given the importance of functional and impermeable lymphatic vessels for the clearance of cellular debris, interstitial fluids, and immune cells [15], we further evaluated the association between donation time and endothelium permeability. We found a strong negative correlation between the donation time frame and VE-Cadherin intensity measured by immunofluorescence (r = −0.812, *p* = 0.008) (Figure 3C). Immunofluorescence imaging represented in Figure 3D demonstrates the difference in intercellular junctions (stained by anti-VE-Cadherin antibody) between aHDLEC incubated with CCP donated 27 days after the onset of symptoms (upper panels) and CCP donated 101 days after the onset of symptoms (lower panels). VE-cadherin (*CDH5)* mRNA expression was also strongly correlated with the time of donation (r = −0.725, *p* < 0.0001) (Figure 3E). Moreover, the permeability of the lymphatic endothelium was characterized using ovalbumin diffusing through the endothelium. A large amount of ovalbumin-488 passing through depicts a highly permeable endothelium. The quantity of ovalbumin-488 diffusing through the endothelium was positively correlated to the time between the onset of symptoms and donation (r = 0.690, *p* = 0.013) (Figure 3F). Therefore, early donation should be included in the CCP-selection process given its correlation with decreased permeability, foreseeing a beneficial effect on the lymphatic endothelium integrity.

### 2.5. Late Donations of COVID-19 Convalescent Plasma Contain Elevated Pro-Inflammatory Cytokines Levels

Given the association between a late donation and lymphatic endothelium dysfunction, we hypothesized that convalescent plasma donated at a later stage after the resolution of symptoms could contain more pro-inflammatory cytokines and mediators. Thus, we characterized the pro- and anti-inflammatory profile of cytokines contained in our convalescent plasma samples. We observed a positive correlation between the time of donation and the levels of pro-inflammatory mediators interleukin 1-beta (IL-1β), IL-13, IL-12p70, interferon gamma (IFNɣ), IL-17A, interferon gamma-induced protein 10 (IP-10), macrophage inflammatory protein-1 (MIP-1), IFNα and IL-1α (Figure 4A–I). We noted previously a decrease in endothelial activation following a treatment with convalescent plasma donated in a later stage (Figure 3A,B). Since pro-inflammatory cytokines are increased in late donated plasma, we sought to investigate whether this decrease could be due to another marker contained in convalescent plasma, namely extracellular vesicles (EVs).

### 2.6. Extracellular Vesicles Derived from Human Lymphatic Endothelial Cells Are More Abundant in Early Donated Plasma

EVs are small vesicles released from apoptotic or activated cells. They contain proteins, lipids, and mRNAs also present in their cell of origin. They are also used as prognostic markers for several pro-inflammatory diseases [30]. In our study, we characterized the presence of EVs derived from red blood cells (CD235a^+^), total leucocytes (CD45^+^), LEC (CD45^−^ podoplanin^+^), blood endothelial cells (CD45^−^ CD62e^+^) and platelets (CD45^−^, C-type lectin-like type II + (CLEC2^+^)) in the plasma of COVID-19 convalescent donors (Figure 5A–G). We observed low concentrations of CD45^−^ podoplanin^+^ EVs compared to the rest of the EVs population quantified (Figure 5H). Interestingly, CD45^−^ podoplanin^+^ EVs correlated with the time of donation (r = −0.413, *p* = 0.004) (Figure 5I), whereas other EVs population did not (Appendix A). In multivariate analysis, concentration of total EVs (CFSE^+^) (β = 0.350, *p* = 0.011) and delay since symptoms onset (β = −0.317, *p* = 0.020) were independent predictors of the concentration of CD45^−^ podoplanin^+^ EVs in convalescent plasma. We herein demonstrate that although CD45^−^ podoplanin^+^ EVs represents a minority of EVs in CCP, they are the only population to correlate with the time of donation, a characteristic believed to be beneficial for the lymphatic endothelium.

### 2.7. Elevated Plasma Extracellular Vesicles Derived from Human Lymphatic Endothelial Cells Correlates with Enhanced Lymphatic Endothelium Integrity

Extracellular vesicles are abundant in lymph [31]. Since plasma ultrafiltrates are collected by the lymphatic system after escaping from the bloodstream, we envision that EVs could easily access the lymphatic circulation along with proteins, cells debris and other macromolecules [30,31]. Since LEC-EVs are more abundant in early donated convalescent plasma, we wanted to explore the relation between EVs characterized in the plasma and lymphatic endothelium integrity (Table 2). First, we observed a positive correlation between CD45^−^ podoplanin^+^ EVs and increased early apoptosis as well as decreased permeability, both markers of preservation of the integrity of lymphatic endothelial cells. Furthermore, unlike CD235a^+^, CD45^+^ and CD45^−^ CLEC2^+^ EVs (i.e., red blood cells, leucocytes and platelets EVs, respectively), CD45^−^ podoplanin^+^ levels are positively correlated to the expression markers of the activation of LEC such as *ICAM1* and vascular endothelial growth factor receptor 3 (VEGFR-3). Indeed, we observed a correlation between VEGFR-3 expression and markers of endothelial integrity (Appendix A). Specifically, *FLT4* expression was associated with decreased endothelial permeability (Appendix A), increased podoplanin (*PDPN)* expression (Appendix A) and decreased percentage of MitoSOX^TM^ Red-positive cells (Appendix A).

To distinguish the confounding effects of donation delay since symptoms, antibody titers, tested cytokines and LEC-EVs on lymphatic endothelial function, we performed multiple linear regression. LEC-EVs was the only variable to be independently associated with ICAM-1 (β = 0.776, *p* = 0.001) and VEGFR expression (β = 0.450, *p* = 0.040), as well as with LEC necrosis-to-apoptosis ratio (β = −0.469, *p* = 0.010), MitoSOX^TM^ Red-positivity ratio (β = −0.589, *p* = 0.020) and LEC permeability (β = −0.813, *p* = 0.001). Thus, the abundance of LEC-EVs in the plasma of convalescent donors could be a useful marker indicating that CCP promotes favorable lymphatic function with incubation.

### 2.8. Secretion of Extracellular Vesicles Derived from Human Lymphatic Endothelial Cells Reflects an Alteration of the Lymphatic Endothelium

EVs production occurs during apoptosis or cell activation. We sought to determine the quantity of EVs secreted by the lymphatic endothelium in culture following CCP treatment. We quantified CD45^−^ podoplanin^+^ EVs collected in the supernatant post-treatment by subtracting the quantity measured in the plasma. As expected, we noted an association between human LEC-derived EVs secretion and increased percentage of cells positive for MitoSOX^TM^ Red (r = 0.407, *p* = 0.039) and cells in the sub G_0_/G_1_ phase of the cell cycle (r = 0.8664, *p* < 0.0001) (Figure 6A,B). Cells in the sub-G_0_/G_1_ phase of the cell cycle are known to be apoptotic cells [32]. Likewise, the quantity of LEC-EVs was correlated with a decrease in cells in the G_2_/M phase (r = −0.604, *p* = 0.022) (Figure 6C), which represents active mitosis [33]. Since the expression of VEGFR-3 is associated with the preservation of the integrity of the lymphatic endothelium [28,29,34], we sought to investigate whether LEC-EVs could also be correlated to its expression. Following a quantification of VEGFR-3 expression upon aHDLEC, we found a negative correlation between its expression and the quantity of CD45^−^ podoplanin^+^ EVs contained in the supernatant by RT-qPCR (r = −0.576, *p* = 0.005) and immunoblot (r = −0.424, *p* = 0.039) (Figure 6D–F). Thus, EVs released by the lymphatic endothelium is related to impaired lymphatic endothelium integrity.

## 3. Discussion

The COVID-19 pandemic is affecting an exponentially growing number of people and is still a high-risk disease [1]. Endothelial damage following cytokine storm is experienced by severely infected patients [3,4,5]. Given the impact of a dysfunctional lymphatic system on the evolution of the severity of inflammatory diseases [35,36], treatments should ideally take into consideration the effect on the lymphatic network. Convalescent plasma therapy has been used in various contexts, including for SARS-CoV-1 [37,38], Ebola virus [39] and Lassa fever [40], among others. The safety and efficacy of CCP therapy were also tested by several clinical trials, as summarized in a published review [11]. Most of these studies confirmed the safety of that procedure. However, the efficacy of this treatment to reduce mortality was mitigated. In particular, the CONCOR-1 study, an international randomized controlled trial, aimed to assess the efficacy of this therapy using mortality and intubation of patients at day 30 as the primary endpoint [10]. The trial, and others as reviewed elsewhere [18], selected the CCP based on the titer of anti-SARS-CoV-2 antibodies to ultimately transfuse it to infected patients [10]. Ultimately, this study failed to show the superiority of CCP over standard of care and even raised the possibility of harmful effects in plasma with low antibody content [10]. The mechanism by which plasma could have a deleterious effect on COVID-19 is unclear. It may be that the strategy of selection of plasma to be injected in ill individuals needs to be optimized based on physiological parameters other than the anti-SARS-CoV-2 antibodies. In fact, CCP with low antibody titers are more likely to have been collected later after the onset of symptoms and therefore to contain less LEC-EVs, which could be a confounding factor. The lymphatic system is a key player in the clearance of immune cells and pro-inflammatory modulators [15]. As vascular complications are a key pathogenic mechanism in COVID-19, leading to inflammation and multiple organ failure, we believe that sustaining lymphatic vessel function should be a factor to be considered to define optimal convalescent plasma therapy. In this study, we identified a particular signature of CCP that would counteract the effects of an excessive inflammation on the lymphatic endothelium. We herein demonstrate that an early donation of convalescent plasma correlates with more plasma LEC-EVs and could protect against the loss of LEC integrity observed during inflammation.

We first assessed the association between the antibody concentrations and clinical parameters, since level of antibodies is the most used CCP selection parameter for clinical trials [10,18]. We observed that patients experiencing a longer duration of symptoms have more anti-SARS-CoV-2 antibodies. Multivariate analysis suggested this may be because these patients also experienced more severe symptoms, which are known to correlate with higher antibody concentration. We did however observe an independent trend for decreased antibody concentration over time since symptoms onset, which is in line with the literature. We then found that aHDLEC exposed to late donations of CCP expressed increased mitochondrial ROS production and lymphatic endothelium permeability, which could potentially be harmful. Increased ROS production is harmful for cells, as it can lead to oxidative stress and cell death [20,21]. Furthermore, one of the key roles of the lymphatic system is to drain the excess fluids of the interstitial tissues, containing cellular debris, immune cells and pro-inflammatory mediators [15]. A disrupted lymphatic endothelium would impair the clearance of interstitial fluids and promote the accumulation of immune cells and pro-inflammatory mediators, which may lead to organ failure [15,41].

In line with studies assessing the correlation between the severity of the disease and benefits of CPT [22,24], we observed that patients experiencing stronger symptoms have more anti-SARS-CoV-2 antibodies, are older and have a longer duration of illness. We also observed that severe symptoms experienced by donors were correlated with more necrosis and less early apoptosis of aHDLEC. Early apoptosis is a defense mechanism initiated by cells to avoid or limit cellular damage [42]. It is also responsible for promoting cell death without causing further inflammation and damage [43]. This is not the case with necrosis, as it induces greater inflammation, proven to be harmful to cells surrounding the dead cell [44]. This inflammation could be driven by the pro-inflammatory mediators contained in the severe CCP. Indeed, Rauch et al. demonstrated endotheliopathy induced by plasma from critically ill patients and noticed cytotoxicity as soon as one hour after the incubation on endothelial pulmonary cells [4]. The observed cytotoxicity was due to biomarkers or the inflammation reaction per se. In light of these results, selection of CCP based on the severity of symptoms experienced by donors may be questionable, since it is shown to be detrimental for the lymphatic endothelium by promoting necrosis. Further, in multiple studies, including ours, the severity of symptoms was assessed by donors using a self-questionnaire, which may introduce a bias in this parameter. Being aware of this limitation, this parameter is deemed unreliable to be taken into consideration for convalescent plasma selection.

Several clinical trials studied the time of administration of CCP to optimize remission [18,22,45]. The study by Beaudoin-Bussières et al. also proposed that early donation should be favored compared to late donation, given the decrease in anti-SARS-CoV-2 antibodies over time [26]. However, a recent study highlighted the lack of efficacy of the early administration of CCP to high-risk patients since it did not prevent the progression of the disease [46]. Given the mitigated opinion upon the efficacy of an early administration of CCP, we hypothesized that the timeframe of selection of CCP, not the time when it is given to patients, should be considered. We observed that early donation correlated with an activation of the lymphatic endothelium, previously found to be essential for rapid migration of T cells to draining lymph nodes, an important mechanism for the resolution of infectious diseases [47]. Furthermore, in this study, an early donation of CCP was also correlated with a preservation of LEC junctions. An adequate permeability of initial lymphatics is crucial for tissue homeostasis and lymphatic function [48]. Increased permeability could promote lymphatic dysfunction and lead to inflammation and edema [48]. In the study by Cromer et al., they show an increased lymphatic permeability, characterized by a decrease in VE-cadherin expression, following a cytokine treatment [48]. Another study confirmed the importance of VE-cadherin expression in lymphatic permeability, as the inhibition of VE-cadherin led to increased permeability [49]. Thus, the decreased permeability of aHDLEC observed in our study suggests a beneficial impact of early donated CCP administration. Another study identified the need to focus on the donation window and highlighted a 60-day post onset of symptoms period for high-titer anti-spike protein CCP [50]. Herein, we describe an additional potential reason to select plasma early following the resolution of symptoms. In this regard, clinical trials may need to consider timing of plasma collection independently of antibody titer since plasma collected late after disease resolution could still be associated with LEC injury despite having high antibody titers.

Bonny et al. quantified cytokine and chemokine levels in CCP and demonstrated higher levels of IFNγ, IL-10, IL-15, IL-21, and MCP-1 compared to plasma of healthy donors [51]. Since the late-donated CCP appears to be more detrimental for the endothelium, we hypothesized that a late donation of CCP would contain more pro-inflammatory mediators than an early donated CCP. We quantified cytokines contained in the convalescent plasma and correlated their levels with the time of donation. A late donation was correlated with higher pro-inflammatory cytokines levels, which could explain the detrimental impact observed on the lymphatic endothelium.

EVs are shown to be able to modulate and regulate the activation of the endothelium by increasing the expression of ICAM-1 [52]. Additionally, Rosell et al. reports that patients infected with COVID-19 have elevated circulating EVs tissue factor activity which is associated with severity of the disease [53]. Krishnamachary et al. also found correlations between severity of the disease and EVs-associated proteins as well as EVs-mediated endothelial apoptosis [54]. EVs can also serve as decoys for neutralizing antibodies and modulate immunity [55]. A review article from Askenase focuses on the involvement of exosomes in convalescent plasma for COVID-19 therapy [56]. Thus, we characterized the EVs population contained in the plasma, focusing on EVs derived from total leucocytes, platelets, erythrocytes as well as blood and lymphatic endothelial cells. In this study, we highlighted an elevated concentration of CD45^−^ podoplanin^+^ EVs (i.e., LEC-EVs) in early donated plasma, whereas EVs levels from leucocytes, blood endothelial cells, platelets and erythrocytes remained unchanged depending on the time of donation.

We then sought to determine the reason why LEC-EVs are more abundant in early donated plasma. We investigated the impact of LEC-EVs contained in the plasma on a lymphatic endothelium. LEC-EVs correlated with an increase in early apoptosis, but most importantly, a decrease in lymphatic endothelium permeability, as indicated by less ovalbumin diffusing through the endothelium. In several studies, EVs were identified as being used as defense mechanisms for sending alarm signals to target cells [57,58]. They are also present in lymph [31] and can transport microRNA, lipids, proteins and growth factors, among others, and be internalized by lymphatic endothelial cells [30]. EVs can release their content and modify the cellular metabolism and function of targeted cells [57]. In our study, cells incubated with plasma containing high levels of LEC-EVs expressed higher levels of *FLT4* and *ICAM1*, demonstrating a lymphatic endothelium activation following incubation of LEC-EVs-rich CCP.

Given the beneficial impact observed of LEC-EVs contained in the plasma on aHDLEC in culture, we hypothesized that these EVs were secreted by the dysfunctional and ill lymphatic endothelial cells of the donors in the attempt to preserve the surrounding lymphatic cells. To validate this, we examined the state of aHDLEC and correlated their condition to the quantity of EVs secreted in culture. The results observed seemed to corroborate our assumptions. Indeed, aHDLEC in culture secreting the highest amount of EVs were those approaching an inflammatory and apoptotic state, as characterized, respectively, by an increase in mitochondrial ROS production and a sub-G_0_/G_1_ phenotype of the cell cycle. Finally, aHDLEC secreting the most amount of EVs were associated with less expression of VEGFR-3, emphasizing the altered state of the lymphatic endothelium. Indeed, in pro-inflammatory conditions, VEGFR-3 is deemed beneficial by promoting lymphangiogenesis and increasing lymphatic contraction capacity, thus limiting inflammation [13,28,29,34,59].

## 4. Materials and Methods

### 4.1. Collection of COVID-19 Convalescent Plasma

Plasma from 45 convalescent donors was collected by apheresis [10]. Donors were required to have resolution of symptoms at least 14 days prior to donation and to have been confirmed positive for COVID-19 by PCR testing. The severity of the symptoms was obtained through a questionnaire filled by donors (appended document). Eligible men or women donors (women with no history of pregnancy) gave consent for a plasma donation of 500–700 mL, which was separated into aliquots and frozen at −80 °C. The collection and distribution of the samples was performed and administered by Héma-Québec, an organization that collaborates with several teams studying convalescent plasma, including the team responsible for the pan-Canadian clinical study CONCOR-1. The research project was approved by the Montreal Heart Institute Ethics Committee (protocol #2021-2812) and Héma-Québec (protocol #2020-004) and in accordance with the Declaration of Helsinki.

### 4.2. Cell Culture

Primary adult human dermal lymphatic endothelial cells (aHDLEC, PromoCell, cat. C-12217) were used in this study and called adult human lymphatic endothelial cells (aHDLEC) throughout the manuscript. The cells were cultured with EGM-MV-2 (supplemented by FCS-25, hEGF-2.5, HC-100, VEGF-0.25, hbFGF-5, R3 IGF-1, AA-500) complete culture medium (PromoCell, cat. C-39221). Heparin (Stemcell Technologies, cat. 07980) at a final concentration of 8 μg/mL was added to basal medium EBM-MV-2 (PromoCell, cat. C-22221) (depleted of fetal bovine serum (FBS) or growth factors) to prevent clotting of the CCP. Antibiotics (Penicillin-Streptomycin, ThermoFisher Scientific, Waltham, Massachusetts, USA, cat. 15140148) at a final concentration of 1%, to ensure inhibition of bacterial growth, were also added to the medium. The cells were used at a passage between 4 and 6, when a confluence of at least 80% was reached. The experimental design of this study is illustrated in Appendix A. Adult HDLEC were treated with 10% convalescent plasma or control plasma for 4 h in EBM-MV-2 and heparin. The control plasma was a commercial human plasma depleted of fibrinogen and treated according to FDA recommendations (SeraConTM II Negative Diluent) and free of any pathogens, including SARS-CoV-2. To mimic the cytokine storm observed in individuals infected with SARS-CoV-2, we added interleukin 6 (IL-6) (PeproTech, cat. 10778-280), tumor necrosis factor alpha (TNFα) (R&D Systems, cat. 210-TA-10), and interferon gamma (IFNɣ) (PeproTech, cat. 10773-476) at final concentrations of 20 ng/mL, 20 ng/mL, and 10 ng/mL, respectively for 20 h. Two additional controls were used in the experiments. The first was an inflamed control, to which cells were incubated with basal medium for 4 h, followed by the cytokines for 20 h. The second one was a healthy control to which the cells were incubated with complete medium for 24 h.

### 4.3. Production of Reactive Oxygen Species

For detection of mitochondrial ROS production, aHDLEC in 6-well plates following treatment (*n* = 26) were detached with accutase (Sigma, cat. A6964), centrifuged and transferred to a 96-well plate (Sarstedt, cat. 82.1583). Following centrifugation, aHDLEC were labeled with MitoSOX^TM^ Red probe for 30 min at 37 °C. All samples were analyzed on a flow cytometer (BD FACSCelesta^TM^). The results obtained were analyzed using FlowJo^TM^ version 10 software (Tree star). The gating strategy is shown in Appendix A. Information on the MitoSOX^TM^ Red probe is described in Appendix A. Hydrogen peroxide (H_2_O_2_, Sigma, cat. H1009) at 1 mM was used as a positive control for the experiment (Appendix A). All CCP data were normalized to the control plasma.

### 4.4. Measurement of Cell Viability

For the measurement of cell viability, treated aHDLEC (*n* = 26) were centrifuged, resuspended in cold Annexin V buffer 1X (BD BioSciences, cat. 5-66121E) and transferred to a 96-well plate. Following centrifugation, cells were labeled with Annexin V and propidium iodide (PI) for 15 min at room temperature (RT) and cold Annexin V buffer 1X was added. All samples were analyzed in flow cytometry (BD FACSCelesta^TM^). The results obtained were analyzed using FlowJo^TM^ version 10 software. The gating strategy is shown in Appendix A. Information on Annexin V and PI is described in Appendix A. Staurosporine (Cayman, cat. 81590) at 1 μM was used as a positive control for apoptosis in the experiment (Appendix A). All data were normalized to the control plasma.

### 4.5. Assessment of Cell-Cycle Distribution

Adult HDLEC were fixed in cold 70% ethanol, washed and stained with a propidium iodide (PI) solution (Biotium cat. #40017), which is fluorogenic and binds to nucleic acids in a stoichiometric manner to allow for the assessment of the proportion of cells in each phase of the cell cycle (pre-replicative (G_0_/G_1_), replicative (S) and post-replicative and mitotic cells (G_2_/M)) [60]. The results obtained were analyzed using ModFit LT^TM^ version 5.0 software (Verity Software House, Topsham, Maine, USA). All CCP data (*n* = 14) were normalized to the control plasma. Information about the PI solution is described in Appendix A.

### 4.6. Quantification of Receptor-Binding-Domain Antibodies

Antibodies within convalescent plasma were determined by human anti-SARS-CoV-2 S RBD ELISA as previously described [61]. Briefly, recombinant SARS-CoV-2 S RBD protein was prepared [26] and adsorbed (2.5 μg/mL) to plates (Immulon 2 HB, ThermoFisher Scientific) overnight at 4 °C. Coated wells were blocked with a buffer (phosphate-buffered saline (PBS) containing 0.1% Tween 20 and 2% BSA) for 1 h at RT. Diluted plasma (1/100) was incubated with the RBD-coated wells for 1 h at RT followed by an incubation with anti-human polyvalent IgA + IgG + IgM (H + L) conjugated to horseradish peroxidase (HRP) (diluted in blocking buffer) for 1 h at RT. The activity of HRP enzyme was assessed following the addition of 3,3′,5,5′-Tetramethylbenzidine (TMB, ESBE Scientific, Saint Laurent, Canada). The colorimetric reaction proceeded for 20 min at RT and was stopped by addition of H2SO4 1N (ThermoFisher Scientific). The plates were then read within 30 min at 450 nm using a Synergy H1 microplate reader (BioTek).

### 4.7. Quantification of Plasma Cytokines

Pro- and anti-inflammatory mediators within 41 convalescent plasmas were quantified according to the manufacturer’s protocol (Inflammation 20-plex Human ProcartaPlex Panel, ThermoFisher Scientific, cat. EPX200-12185-901).

### 4.8. Analyses of Extracellular Vesicles in Plasma and Lymphatic Endothelial Cell Supernatant

For the staining of EVs in plasma (*n* = 48), 10 μL of CCP or control plasma were added to 90 μL of Annexin V buffer 1X (BD BioSciences, cat. 5-66121E) supplemented with 10 μM D-phenylalanyl-L-prolyl-L-arginine chloromethyl ketone (PPACK) (Cayman, cat. 15160-1). The antibody mix including carboxyfluorescein succinimidyl ester (CFSE) to establish the presence of active esterases and antibodies binding major histocompatibility complex 1 (MHC-I), CD235a, CD45, C-type lectin-like type II (CLEC2), CD62e, and podoplanin with the concentrations described in Appendix A was then performed in Annexin V buffer 1X and filtered through a column (Millipore, cat. UFC500396) centrifuged at 10,000g for 1 min, and 100 μL of the antibody mix was added to the plasma [62].

Stained EVs in CCP and in the supernatant (*n* = 26) of treated aHDLEC were quantified as described previously [63]. Briefly, samples were processed with a flow cytometer (BD FACSCelesta^TM^) in which the 450/40 bandpass filter (BV421, violet laser) was manually swapped after cytometer setting and tracking (CS and T) calibration with a 1 mm-thick magnetron sputtered 405/10 bandpass filter (Chroma Technology, Bellows Falls, VT, USA), referred to as V-SSC in this manuscript. Plots and histograms show all parameters in height (indicated as –H), as recommended for EV detection [64]. Events were acquired at a flow rate of 12 µL/min, which is the lowest flow rate on the FACSCelesta. The flow rate during acquisition was kept to a minimum to avoid swarming effects and coincident detection [65]. For a more precise calibration for assessment of biological vesicle size (refractive index in the range 1.36 to 1.42), the flow cytometer was calibrated for EV detection using the ApogeeMix (#1493, Apogee Flow Systems, Hemel Hempstead, UK), which consists of a mixture of non-fluorescent silica beads (180, 240, 300, 590, 880, and 1300 nm) and FITC-fluorescent latex beads (110 and 500 nm) (Appendix A). The EV gate was set to contain events ranging from ~100 to ~1000 nm using the size of the non-fluorescent silica beads in the ApogeeMix, whose refractive index is close to that of cellular membranes [66]. The threshold for the forward scatter (FSC) detector was set at the lowest possible value (200 V) in FACS Diva software (BD Biosciences). The background in the EV gate was determined by running samples containing all reagents and antibodies except EV-containing culture media and was subtracted from the values obtained for samples with EV-containing culture media. Single-labeled tubes and one unlabeled tube were also analyzed to adjust the cytometer voltages, compensations, and to create the analysis strategy shown in Appendix A. To confirm the cellular origin of the extracellular vesicles detected, 0.1% Triton X-100 (0.05% final concentration) was added to the samples for 30 min, and the decrease in EVs count was denoted (Appendix A). Data were analyzed using FlowJo^TM^ software (Tree Star Inc., Ashland, Oregon, USA).

For characterization of EVs in the supernatant of aHDLEC, 50 μL was mixed with 50 μL of Annexin V buffer 1X and all subsequent steps were identical to those for plasma. We characterized the composition of convalescent plasma in EVs by using the gating strategy shown in Appendix A and Figure 5A–G. The absolute concentration of EVs/mL was calculated using count beads (Apogee Flow System, cat. 1426) and Equation (1), *cb*, count beads; *V*, volume; *DF*, dilution factor; *Bg*, background sample.
(1)$$\left[EVs\right]=\;\left(\frac{\#EVs}{\#cb}\;\times \;\frac{Vcb\;\times \;\left[initial\;cb\right]}{Vtotal}\;\times DF\right)\;\times 1000-\;\left[EVs\;in\;Bg\right]$$

### 4.9. Total Cholesterol

Measurement of cholesterol in CCP samples (*n* = 33) was performed according to the manufacturer’s protocol (Cholesterol E-Test, FUJIFILM Wako Pure Chemical Corporation, cat. 439-17501).

### 4.10. Messenger RNA Analysis by RT-qPCR

Treated aHDLEC were harvested, suspended in RiboZol™ RNA Extraction Reagent and stored at −80 °C for at least 24 h. The RNA was extracted using the PureLink RNA Mini Kit extraction kit (Invitrogen, cat. 12183025) according to the manufacturer’s protocol and quantified on a NanoDrop™ 1000 Spectrophotometer (ThermoFisher Scientific). Reverse transcription of RNA was performed using the High-Capacity cDNA Reverse Transcription Kit (ThermoFisher Scientific, cat. 4368814). Quantitative PCR was performed on the QuantStudio™ 3 (ThermoFisher Scientific) using 10 ng of complementary DNA (cDNA) mixed with Itaq™ Universal SYBR^®^ Green Supermix (Biorad, cat. 1725121). The primers used are displayed in Appendix A. The amplification cycles were performed at 94 °C for 10 s and at 60 °C for 45 s for 40 cycles. The relative expression was calculated by the comparative method of (2^−ΔΔCT^) and normalized to the housekeeping gene *ACTB*. All CCP samples were then normalized to the control plasma.

### 4.11. Immunofluorescence

Adult HDLEC were seeded in 8-well plates (Sigma, cat. PEZGS0816) at 30,000 cells per well until confluence was reached. Following treatment, aHDLEC were incubated with 10 ng/mL Wheat Germ Agglutinin, Alexa Fluor^TM^ 488 Conjugate (ThermoFisher Scientific, cat. W11261) for 7.5 min in the dark and fixed in PFA 2% for 25 min. Cells were then blocked in 2% Donkey Serum for 30 min at RT and incubated with anti-VE-Cadherin antibody (Abcam, cat. Ab33168) for 1 h at RT at a final dilution of 1/200. Secondary antibody Alexa Fluor^TM^ anti-rabbit 647 (Jackson ImmunoResearch, cat. 711-606-152) was incubated for 1 h at RT at a final dilution of 1/300. Adult HDLEC were then incubated with DAPI (1/250) for 30 min and visualized using an LSM 710 Confocal Microscope (Zeiss) equipped with a x63/1.4 oil dic objective. Images were analyzed using ImageJ software and relative intensity was obtained by normalization of CCP samples (*n* = 9) to the control plasma.

### 4.12. Transwell Permeability Assay

Adult HDLEC were seeded in transwell inserts (Corning cat. 3470 and Greiner Bio-one cat. 662640) in a 24-well plate (vWR cat. 10861-700) and treated in the apical side. Inflamed and healthy aHDLEC were used as positive (incubated with basal medium and cytokines) and negative control (incubated with complete medium), respectively. Permeability was assessed by the addition of ovalbumin Alexa Fluor^TM^ 488 (50 ng/mL) (Invitrogen cat. 34781) in a phenol-free medium (Promocell, cat. C-22226) for 20 min at RT followed by its recovery in the basolateral region. The medium containing the ovalbumin was transferred in a 96-well black flat plate and its fluorescence was read with the plate reader Synergy 2 (BioTek) with 485/20 nm of excitation and 528/20 nm of emission. All CCP samples (*n* = 12) were then normalized to the control plasma.

### 4.13. Immunoblotting

Proteins were extracted from treated aHDLEC using ice-cold radioimmunoprecipitation (RIPA) assay buffer and the protein concentrations were determined using MicroBCA™ Protein Assay Kit (ThermoFisher Scientific, cat. 23235). Protein samples were diluted in 4× Laemmli buffer, heated at 95 °C for 5 min and separated by electrophoresis on a 7.5% SDS-PAGE. Proteins were then transferred on a poly (vinylidene fluoride) (PVDF) membrane overnight at 4 °C. The membranes were blocked with 5% nonfat dry milk or 5% bovine serum albumin (BSA) in Tris-buffered saline (TBST, 0.1% Tween 20) for 2 h at RT, then incubated with an anti-VE-Cadherin (Abcam, cat. Ab33168), anti-phospho-VE-Cadherin (Invitrogen, cat. 44-11446), anti-VEGFR-3 (ProteinTech, cat. 20712-1-AP) and anti-α/ß tubulin (Cell Signaling, cat. 2148) overnight at 4 °C. The membranes were incubated with HRP-conjugated secondary anti-rabbit (Abcam, cat. Ab6721) and anti-mouse (ThermoFisher Scientific, cat. PR-W4021) antibodies for 1 h at RT. An ECL Blotting Substrates Kit (ThermoFisher Scientific, cat. PI32209) was used for detection. Each plasma sample (*n* = 24–26) was normalized with its respective non-phosphorylated or α/ß-tubulin expression followed by a normalization on the control plasma expression.

### 4.14. Statistical Analysis

Normally distributed data are represented as mean and standard deviation (SD), whereas non-normally distributed data are represented as median and interquartile range. For continuous variables passing the Shapiro–Wilk normality test, a two-tailed Pearson’s correlation was performed. For variables that did not pass the normality test, a logarithmic or square-root transformation (if the previous one failed to normalize the data) was performed to obtain a homogeneous distribution. For the variables that still did not pass the normality test following the transformation, the initial values were used, but a non-parametric two-tailed Spearman’s correlation was performed. Finally, in Figure 5H, a Kruskal–Wallis test with a Dunn post hoc test was used. Multiple linear regression was performed using forward method, whenever more than one variable from the univariate analysis was associated with the tested outcome. Reported *p* values are two-sided and the significance level was *p* < 0.05 for all statistical tests. Analyses were performed using SPSS version 27 software and figures were represented using Prism version 9 software (GraphPad).

## 5. Conclusions

In conclusion, we reported in this study a new and easily measurable parameter to take into consideration for the selection of CCP used in CPT. Our results demonstrate that an early donation of convalescent plasma could protect against lymphatic dysfunction, as it contains circulating LEC-EVs. We envision that the secretion of LEC-EVs by an impaired endothelium could be an alarm signal that instigates the self-defense of peripheral lymphatic vessels against an excessive inflammation. Contrariwise, a late donation could promote loss of endothelial integrity and increased permeability of the lymphatic endothelium. Dysfunctional lymphatic drainage could alter the clearance of immune cells and pro-inflammatory mediators, leading to the exacerbation of inflammation in infected COVID-19 patients. Accordingly, an easy and efficient selection of convalescent plasma based on time of donation would be essential to promote the preservation of the lymphatic and immune system of infected patients. Further clinical trials are needed to investigate whether early donations of CCP would be beneficial for the well-being of infected and hospitalized patients.

In terms of limitations, the recruitment of patients occurred solely during the first half-year of the pandemic, which prevented a thorough and longitudinal follow up of donors, and excluded the effect of vaccine. A restricted amount of CCP donors is also another limitation of this study. The severity of the disease was based on self-assessment, and thus graded by donors themselves using a self-questionnaire. Clinical parameters that could have been insightful for the analysis of this study, such as body mass index, history of cardiac events and medications, were also unavailable. Finally, the total count of EVs produced by aHDLEC in vitro was calculated based on the total number found in the media minus the number of EVs contained in the plasma that was incubated on the lymphatic endothelial cell monolayer.

## Figures and Tables

**Figure 1 pharmaceuticals-15-00365-f001:**
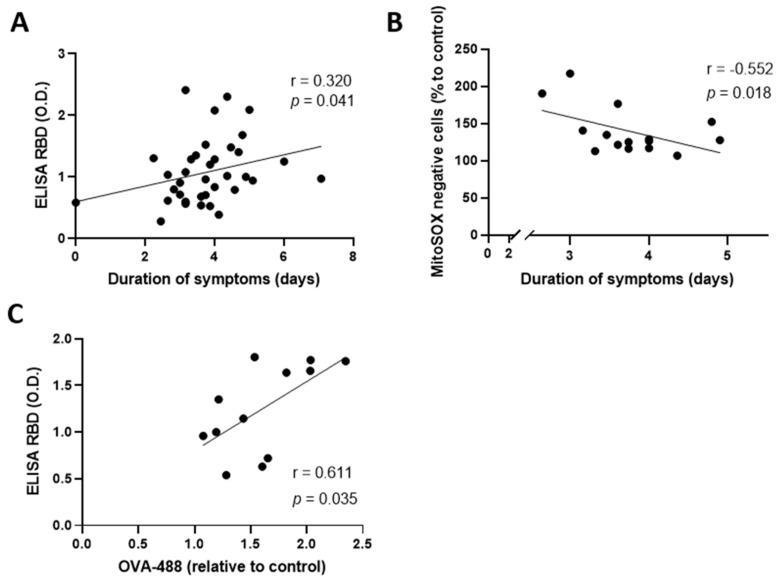
Elevated antibody concentrations and prolonged symptoms are detrimental for the lymphatic endothelium integrity. (**A**) Correlation between the duration of symptoms (square-root transformation) and the concentration of plasmatic SARS-CoV-2- receptor-binding-domain antibodies measured by ELISA. (**B**) Correlation between the duration of symptoms and MitoSOX^TM^ Red-negative cells measured by flow cytometry after incubating aHDLEC with convalescent plasma for 4 h and cytokines cocktail for 20 h. (**C**) Correlation between the concentration of plasmatic RBD antibodies measured by ELISA and the permeability of the endothelium measured by spectrophotometry (absorbance of OVA-488) after incubating human LEC with convalescent plasma for 4 h and cytokines cocktail for 20 h. Each point represents a treatment. Significance (*p* < 0.05) was determined by a Pearson correlation. A square-root transformation of the duration of symptoms to reach normal distribution was performed. RBD—receptor-binding domain; O.D—optical density; OVA—ovalbumin; aHDLEC—adult human dermal lymphatic endothelial cells.

**Figure 2 pharmaceuticals-15-00365-f002:**
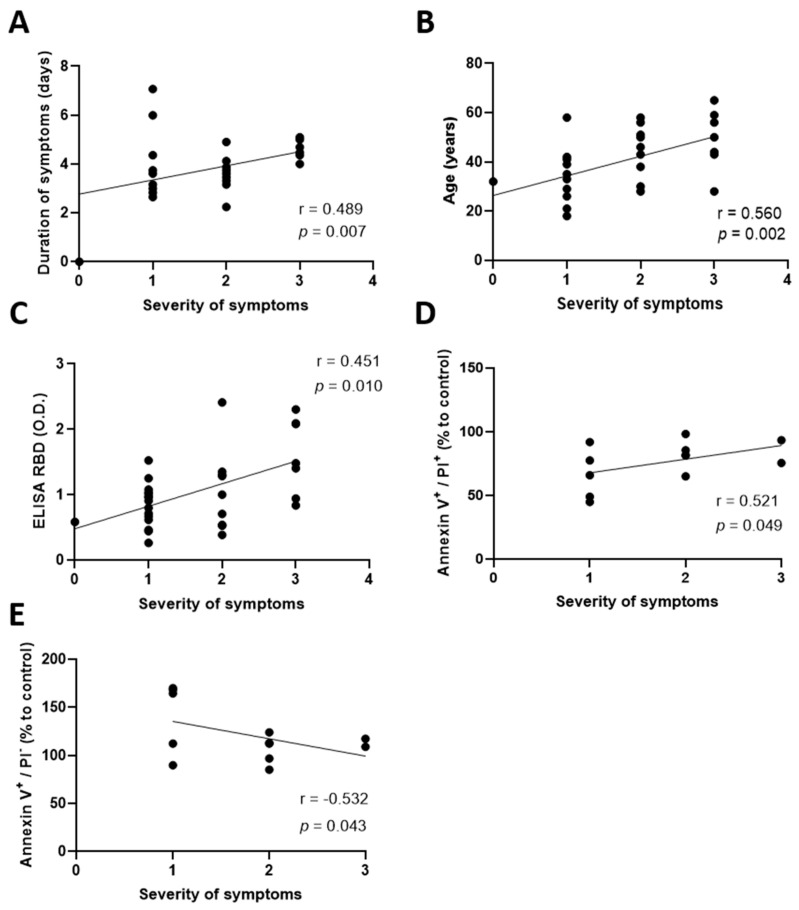
The severity of symptoms correlates with the duration of symptoms, age, receptor-binding-domain antibodies of donors and cell viability of lymphatic endothelial cells. (**A**) Severity was quantified using a questionnaire answered by donors and graded as follows: asymptomatic, mild, moderate and severe. Severe symptoms were correlated with a higher duration of symptoms (square-root transformation to reach a normal distribution). (**B**) Correlations between severity of the symptoms and the age of the donors were performed. (**C**) Severity was correlated with the concentration of antibodies measured by ELISA. (**D**,**E**) Treated aHDLEC were labeled with Annexin V and PI and analyzed by flow cytometry. The severity of the symptoms was correlated with cells in late apoptosis/necrosis (**D**) and in early apoptosis (**E**) represented by Annexin V- and PI-positive cells and Annexin V-positive and PI-negative cells, respectively. Cells are given as percentages relative to cells treated with control plasma. Significance was determined by a Spearman correlation. *p* < 0.05 was considered significant. RBD, receptor-binding domain; O.D, optical density; PI, propidium iodide; aHDLEC, adult human dermal lymphatic endothelial cells.

**Figure 3 pharmaceuticals-15-00365-f003:**
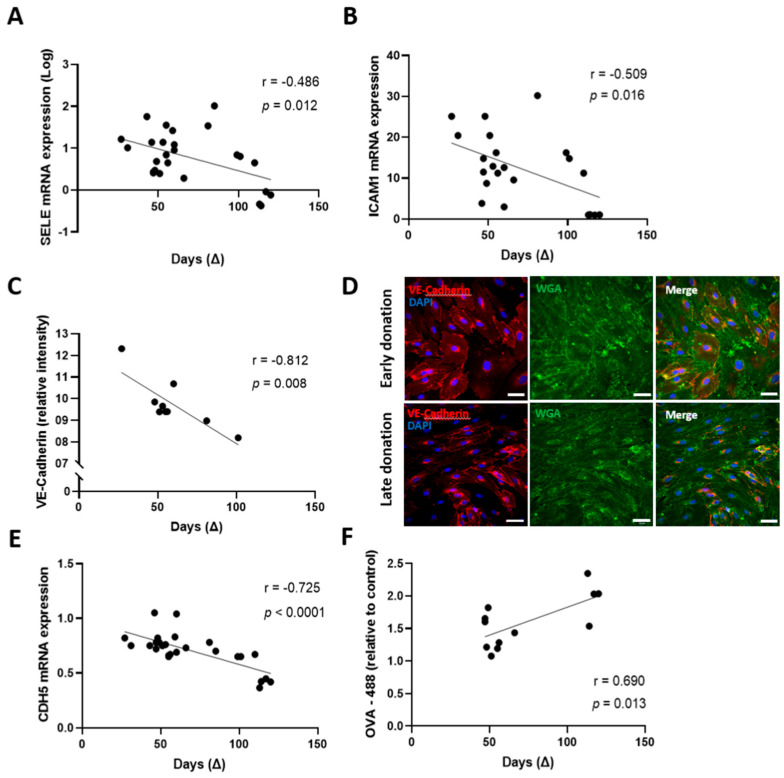
Early donation of COVID-19 convalescent plasma is a good predictor of preserved endothelial integrity. (**A**,**B**) Treated aHDLEC (convalescent plasma for 4 h and cytokines for 20 h) were harvested and the mRNA expression, measured by RT-qPCR, of *SELE* (**A**) and *ICAM1* (**B**) was correlated with the time of donation since the onset of symptoms. (**C**) Following treatment, cells were fixed in paraformaldehyde (PFA) 2% and incubated with anti-VE-Cadherin antibodies. The intensity of VE-Cadherin relative to cells treated with control plasma was correlated to the duration between the onset of symptoms and the donation. (**D**) Immunofluorescence images of treated aHDLEC incubated with CCP donated at 27 days (upper panels) and 101 days (lower panels) post onset of symptoms. The left panels show the expression of VE-Cadherin and DAPI, whereas the middle panel represents the WGA staining. The right panels show the representation of the merged staining. Scale bar = 50 μm. (**E**) VE-Cadherin mRNA expression was correlated to the duration between the onset of symptoms and donation (∆). (**F**) Endothelial permeability was analyzed by the relative concentration of ovalbumin-488 measured following migration through the endothelium compared to control. The concentration of OVA-488 was correlated with the duration between the onset of symptoms and the donation. Significance was determined by Pearson correlation and *p* < 0.05 was considered significant. ∆, time of donation since onset of symptoms; *SELE*, gene coding for E-selectin; *ICAM1*, gene coding for intercellular adhesion molecule 1; WGA, wheat germ agglutinin; *CDH5*, gene coding for VE-Cadherin; OVA, ovalbumin; aHDLEC, adult human dermal lymphatic endothelial cells; CCP, COVID-19 convalescent plasma.

**Figure 4 pharmaceuticals-15-00365-f004:**
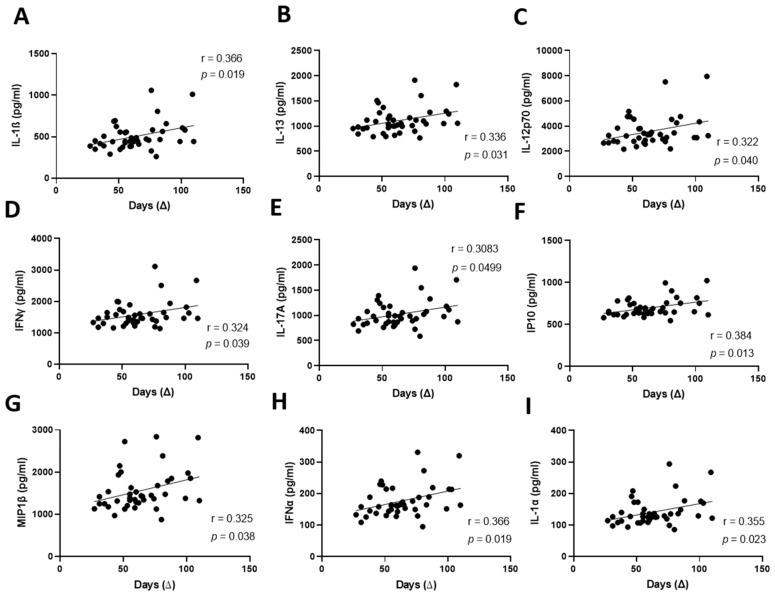
Late donation correlates with higher concentration of circulating pro-inflammatory cytokines. (**A**–**I**)**.** Concentration of pro-inflammatory cytokines and mediators contained in the convalescent plasma was analyzed using a Multiplex kit. The duration between onset of symptoms and donation (∆) was correlated to interleukin 1 beta (IL-1β) (**A**), IL-13 (**B**), IL-12 p70 (**C**), interferon gamma (IFNγ) (**D**), IL17A (**E**), IP-10 (**F**), MIP1 (**G**), IFNα (**H**) and IL-1α (**I**). Significance was determined by a Pearson correlation and *p* < 0.05 was considered significant. ∆, time of donation since onset of symptoms. IL, interleukin; IFN, interferon; IP10, interferon gamma-induced protein 10; MIP1, macrophage inflammatory protein 1.

**Figure 5 pharmaceuticals-15-00365-f005:**
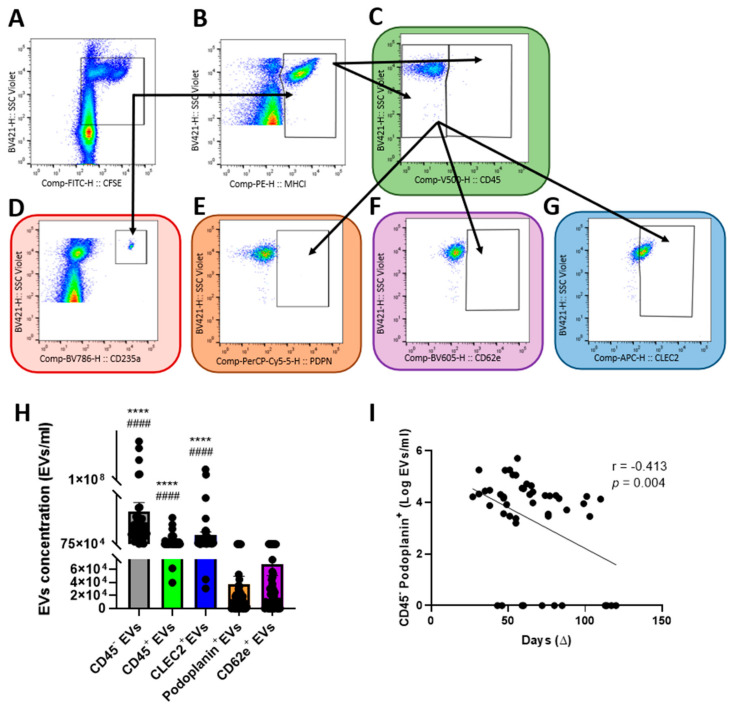
Characterization of extracellular vesicles and correlation with duration between onset of symptoms and donation. (**A**–**G**) EVs gating strategy (side scatter) for CFSE^+^ EVs (**A**), MHCI^+^ EVs (**B**), CD45^+^ EVs (**C**), CD235^+^ EVs (**D**), CD45^−^ podoplanin^+^ EVs (**E**), CD62e^+^ EVs (**F**) and CLEC2^+^ EVs (**G**). (**H**) Concentration of EVs in CCP measured by flow cytometry. Kruskal–Wallis test with a Dunn post hoc test was performed. (**I**) Concentration of CD45^−^ podoplanin^+^ EVs in convalescent plasma and correlation with the duration between the onset of symptoms and the donation. Significance was determined by a Pearson correlation after a logarithmic transformation of CD45^−^ podoplanin^+^ EVs concentration to reach a normal distribution. *p* < 0.05 was considered significant. **** *p* < 0.0001 significantly different from podoplanin^+^ EVs. #### *p* < 0.0001 significantly different from CD62e^+^ EVs. EVs, extracellular vesicles; CCP, COVID-19 convalescent plasma; CFSE, carboxyfluorescein succinimidyl ester; MHCI, major histocompatibility complex I; PDPN, podoplanin; CLEC2, C-type lectin-like type II; ∆, time of donation since onset of symptoms.

**Figure 6 pharmaceuticals-15-00365-f006:**
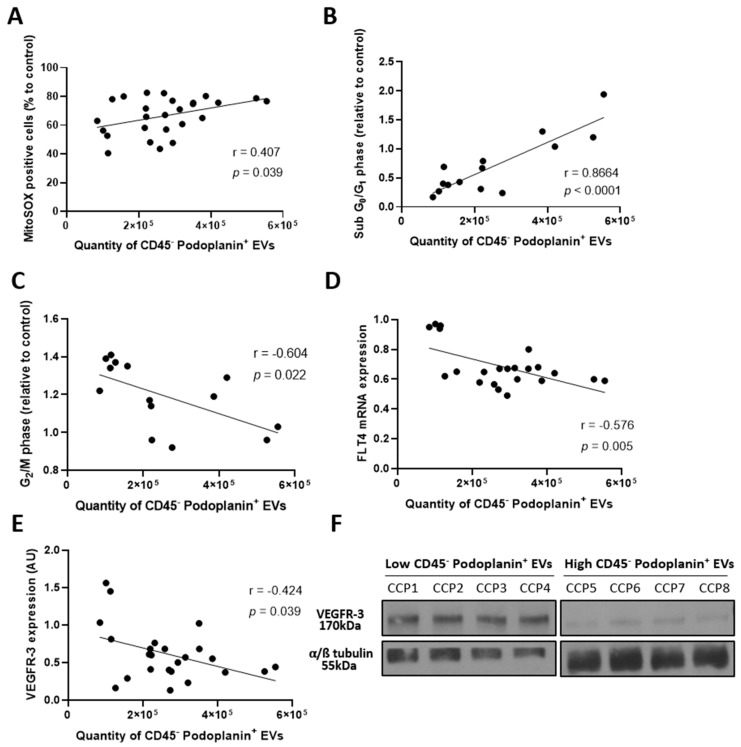
Secretion of lymphatic endothelial cells derived extracellular vesicles is a marker of impaired integrity. (**A**) Treated aHDLEC (convalescent plasma for 4 h and cytokines for 20 h) labeled by the MitoSOX^TM^ Red probe and correlated with the quantity of EVs shed by the lymphatic endothelial cells. (**B**,**C**) Cells in sub-G_0_/G_1_ (**B**) and G_2_/M (**C**) phases of the cell cycle, determined by cell cycle analysis using PI, were correlated to the quantity of EVs secreted by the lymphatic endothelium. (**D**,**E**). Treated aHDLEC were harvested and expression of *FLT4* was assessed by RT-qPCR (**D**) and immunoblot (**E**) followed by a normalization onto the expression of *ACTB* or α/ß tubulin, respectively. The measured expression of VEGFR-3 was then correlated with the quantity of EVs secreted by the endothelium. (**F**) Immunoblot analysis for VEGFR-3 and the loading control α/ß tubulin. The left panel represents aHDLEC secreting low quantities of CD45^−^ podoplanin^+^ EVs (less than 120,000 EVs) following incubation with CCP, and the right represents aHDLEC secreting high quantities of CD45^−^ podoplanin^+^ EVs (more than 380,000 EVs). Significance was determined by a Pearson correlation and *p* < 0.05 was considered significant. EVs, extracellular vesicles; PI, propidium iodide; CCP, COVID-19 convalescent plasma; VEGFR-3, vascular endothelial growth factor receptor 3; FLT4, gene coding for VEGFR-3; aHDLEC, adult human dermal lymphatic endothelial cells; AU, arbitrary unit.

**Table 1 pharmaceuticals-15-00365-t001:** Clinical parameters of donors of convalescent plasma.

Variables	
Age, years	41 ± 14
Female sex, *n* (%)	17 (38%)
Duration of symptoms, days (IQR)	14 (10–20)
Time of donation, days	67 ± 25
ABO Blood group, *n* (%)	
O	18 (40)
A	15 (33)
B	5 (11)
AB	7 (16)
Rhesus, *n* (%)	
Positive	38 (84)
Negative	7 (16)
Severity, *n* (%)	
Asymptomatic	1 (2)
Mild	12 (27)
Moderate	9 (20)
Severe	7 (16)
Loss of smell/taste, *n* (%)	
Yes	20 (44)
No	7 (16)
Unknown	18 (40)
O.D RBD-antibody concentrations, AU (min, max)	1.08 ± 0.54 (0.26, 2.41)
Total cholesterol, mg/dL	188.5 ± 45.1

Values are given as mean ± standard deviation, unless mentioned otherwise. Duration of symptoms is not normally distributed; its value is given as median and interquartile range (IQR). Severity was graded following a questionnaire answered by donors. O.D, optical density; RBD, receptor-binding domain; AU, arbitrary unit; Min, minimum; Max, maximum.

**Table 2 pharmaceuticals-15-00365-t002:** Correlation between extracellular vesicles and lymphatic endothelial cell markers.

	CD235a^+^ (EVs/mL)		CD45^+^ (EVs/mL)		CD45^−^ CLEC2^+^ (EVs/mL)		CD45^−^ Podopanin^+^ (EVs/mL)	
	Pearson (r)	*p* Value	Pearson (r)	*p* Value	Pearson (r)	*p* Value	Spearman (r)	*p* Value
Early apoptosis	−0.026	0.899	0.464	0.017 *	0.158	0.439	0.561	0.003 **
Permeability	0.019	0.954	−0.596	0.041 *	−0.103	0.749	−0.836	0.0007 ***
*FLT4* mRNA	0.322	0.144	0.078	0.729	0.292	0.187	0.616	0.002 **
*ICAM1* mRNA	0.009	0.970	0.300	0.174	−0.185	0.409	0.689	0.0004 ***

CD235a^+^, CD45^+^ and CD45^−^ CLEC2^+^ EVs were transformed using a logarithmic transformation to reach normal distribution. CD45^−^ podoplanin^+^ EVs did not reach a normal distribution despite logarithmic transformation. Early apoptosis determined by cells positive for Annexin V and negative for propidium iodide (PI)-normalized cells treated with control plasma. Permeability was measured by ovalbumin-488 diffusing through aHDLEC following the transwell assay. Correlations of Pearson (for CD235a^+^, CD45^+^ and CD45^−^ CLEC2^+^ EVs) and Spearman (for CD45^−^ podoplanin^+^ EVs) were used. * *p* < 0.05; ** *p* < 0.01; *** *p* < 0.001. EVs, extracellular vesicles; CLEC2, C-type lectin-like type II; *FLT4*, gene coding for vascular endothelial growth factor receptor 3; *ICAM1*, gene coding for intercellular adhesion molecule 1; aHDLEC, adult human dermal lymphatic endothelial cells.

## Data Availability

Data is contained within the article and Appendix A.

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
