# Peer review of "Use of Early Donated COVID-19 Convalescent Plasma Is Optimal to Preserve the Integrity of Lymphatic Endothelial Cells"

_pharmaceuticals, 2022, doi:10.3390/ph15030365_

Round 1

Reviewer 1 Report

In the presented work of Nada et al. authors analyze the impact of COVID19 convalsecent plasma on function and integrity of lymphatic endothelaial cells. This study is interesting and brings adresses key issues in the potential us of COVID19 convalescent plasma use in SARS-CoV-2 infected patient treatment. Some minor comment/issues:

1) In the material and methods section COVID19 was diagnosed with positive RT-PCR test or epidemiologic contact. How many people were diagnozed based on epidemiological data only? How was the time from infection to apheresis measured? What were the antibody titers of this group?

2) Figrue 1, 3 and 4 legend should be placed just following the figure and not in other parts of the text for better receipt.

3) Patients were assigned to the COVID19 severity grup based on a questionare. What kind of questionaire was it and how exaclty were the symptoms graded?

Author Response

In the presented work of Nada et al. authors analyze the impact of COVID19 convalsecent plasma on function and integrity of lymphatic endothelaial cells. This study is interesting and brings adresses key issues in the potential us of COVID19 convalescent plasma use in SARS-CoV-2 infected patient treatment. Some minor comment/issues:

1) In the material and methods section COVID19 was diagnosed with positive RT-PCR test or epidemiologic contact. How many people were diagnozed based on epidemiological data only? How was the time from infection to apheresis measured? What were the antibody titers of this group?

We thank the reviewer for these comments, and are sorry for the confusion. COVID-19 was indeed diagnosed with RT-PCR test for all of our patients included in our study.  We have modified the sentence accordingly (line 531). Time from infection and antibody titers were specified in table 1. 

2) Figure 1, 3 and 4 legend should be placed just following the figure and not in other parts of the text for better receipt.

We apologize for these discrepancies. We think it is probably something that the Journal itself will have to adjust while publishing the manuscript, but we will definitely make sure legends are always right after the figures.

3) Patients were assigned to the COVID19 severity grup based on a questionare. What kind of questionaire was it and how exaclty were the symptoms graded?

To better depict how severity was determined, we have now appended the questionnaire (please see the attachment).

Reviewer 2 Report

I think this paper is very interesting. But there is a big concern. Severity is an important point in this paper, but in what form did you ask the patient? Since medical systems differ from country to country, the severity of the disease was judged differently, so I think you should state it clearly.

Author Response

I think this paper is very interesting. But there is a big concern. Severity is an important point in this paper, but in what form did you ask the patient? Since medical systems differ from country to country, the severity of the disease was judged differently, so I think you should state it clearly.

We thank the reviewer for this comment. To better depict how severity was determined, we have now appended the questionnaire to the manuscript (please see the attachment).

Reviewer 3 Report

Very thorough and interesting review. No comments

Author Response

Very thorough and interesting review. No comments.

We thank the reviewer for its review of the manuscript.

Round 2

Reviewer 2 Report

I think this revised paper is acceptable.